# Clinical Applications of Genomic Alterations in ATLL: Predictive Markers and Therapeutic Targets

**DOI:** 10.3390/cancers13081801

**Published:** 2021-04-09

**Authors:** Noriaki Yoshida, Hiroaki Miyoshi, Koichi Ohshima

**Affiliations:** 1Department of Pathology, Kurume University School of Medicine, Kurume 830-0011, Japan; miyoshi_hiroaki@med.kurume-u.ac.jp; 2Department of Clinical Studies, Radiation Effects Research Foundation, Hiroshima 732-0815, Japan

**Keywords:** adult T-cell leukemia/lymphoma, genomic alterations, predictive markers, therapeutic targets

## Abstract

**Simple Summary:**

In this review paper, we aim to summarize recent findings of genomic alterations found in adult T-cell leukemia/lymphoma (ATLL), which is an incurable disease induced by a virus; human T-cell leukemia virus type 1 (HTLV-1). Genomic alterations of ATLL have been comprehensively analyzed and the identified alterations and HTLV-1 infection synergistically act for ATLL development. As HTLV-1 is an endemic disease, ATLL frequently occurs in the endemic areas. Current clinicogenomic analyses suggest the existence of regional difference in ATLL pathophysiology. From a clinical perspective, several studies identified alterations that act as predictive markers and that a part of the alterations can be targetable in ATLL. The alterations can be leveraged to improve ATLL prognosis.

**Abstract:**

Adult T-cell leukemia/lymphoma (ATLL) is a peripheral T-cell lymphoma (PTCL) caused by human T-cell leukemia virus type 1 (HTLV-1). Recent comprehensive genomic analyses have revealed the genomic landscape. One of the important findings of genomic alterations in ATLL is that almost all alterations are subclonal, suggesting that therapeutic strategies targeting a genomic alteration will result in partial effects. Among the identified alterations, genes involved in T-cell receptor signaling and immune escape mechanisms, such as *PLCG1*, *CARD11*, and *PD-L1* (also known as *CD274*), are characteristic of ATLL alterations. From a geographic perspective, ATLL patients in Caribbean islands tend to be younger than those in Japan and the landscape differs between the two areas. Additionally, young Japanese ATLL patients frequently have CD28 fusions, compared with unselected Japanese cases. From a clinical perspective, PD-L1 amplification is an independent prognostic factor among every subtype of ATLL case. Recently, genomic analysis using deep sequencing identified a pre-ATLL clone with ATLL-common mutations in HTLV-1 carriers before development, indicating that genomic analysis can stratify cases based on the risks of development and mortality. In addition to genomic alterations, targetable super-enhancers have been identified in ATLL. These data can be leveraged to improve the prognosis of ATLL.

## 1. Introduction

Adult T-cell leukemia/lymphoma (ATLL) is a mature T-cell neoplasm induced by human T-cell leukemia virus type 1 (HTLV-1) [1]. HTLV-1 is endemic in several countries, including Japan and the Caribbean islands and ATLL frequently develops in these regions. In fact, ATLL is the most common lymphoma subtype in the HTLV-1 endemic area (Kyushu) of Japan [2], and is the fourth most common type of peripheral T-cell lymphoma (PTCL) worldwide [3]. Approximately 10 to 20 million people are estimated to be infected by HTLV-1 across the world [4]. Breastfeeding is the main route for transmission of the virus from mother to child [5]. ATLL is classified into four clinical subtypes, based on clinical manifestations [6]. Acute and lymphomatous subtypes are regarded as the aggressive types, while the chronic and smoldering subtypes are recognized as indolent types. Chronic type ATLL is further divided into favorable and unfavorable types based on the clinical findings. Since intensive chemotherapies, including stem cell transplantation (SCT), are recommended for the aggressive and unfavorable chronic subtypes of ATLL [7], a precise diagnosis and classification is warranted before treatment. Alternative therapeutic treatments, including SCT and anti-CCR4 antibody which have been approved for use in Japan, have also been found to improve the prognosis of ATLL [8,9,10]. However, further novel therapeutic strategies are needed as ATLL is still considered an incurable disease. In this review we summarize the genomic alterations of ATLL, focusing on differences between geographic regions and ages at onset. We also present genomic alterations reported by recent studies that are recognized as predictive markers and therapeutic targets. 

## 2. Overview of Genomic Alterations of ATLL

In Japan, the annual incidence rate of ATLL was reported to be approximately from 60 to 130 per 100,000 HTLV-1 carriers with male predominance [11,12,13,14]. Additionally, the lifetime risk of ATLL among HTLV-1 carriers was estimated to be 4–6% for men and 2.6% for women [11,13,15]. Therefore, it is widely recognized that additional events accompanied with HTLV-1 infection are required for ATLL development [14]. Risk factors identified for developing ATLL include high pro-viral load, advanced age, family history of ATLL, first opportunity for HTLV-1 screening during treatment for other diseases and certain human leukocyte antigens alleles [16,17], which also suggests that HTLV infection alone is not sufficient for ATLL development. It is unclear about the life-time risk and the risk factors in the other HTLV-1 endemic areas and this point may be important to understand precise ATLL pathophysiology. Since the median age at diagnosis is approximately 70 in Japan, genomic alterations in HTLV-1-infected cells accumulate over a long period. Recent genomic analyses have revealed several alterations in ATLL [18,19,20,21,22]. A comprehensive study conducted by Kataoka and colleagues revealed these alterations in detail using next-generation sequencing (NGS) [20]. These genomic analyses have found that genomic alterations related to T-cell receptor (TCR)/NF-κB signaling and immune escape mechanism were hallmarks of ATLL genomics. 

## 3. Mutations of Genes Involved in TCR/NF-κB Signaling

Actually, alterations of the genes involved in TCR/NF-κB signaling were identified in over 90% of the ATLL cases analyzed. The alterations involved in TCR signaling are divided into three groups: (i) alterations in genes of the proximal components of TCR signaling (e.g., FYN, *PLCG1*, and *VAV1*); (ii) alterations in genes of more distal TCR signaling, which result in the activation of NF-κB signaling (e.g., *PRKCB* and *CARD11*); (iii) alterations in downstream signaling genes (e.g., *IRF4* and *RHOA*) [23]. Additionally, alterations of *CD28*, which co-stimulates TCR signaling, are frequently found in ATLL cases. Co-occurrence of *PRKCB* and *CARD11* mutations has been significantly correlated, although mutual exclusivity in these alterations related to TCR signaling has not been identified [20]. Although the coexistence of both mutants has been reported to synergistically act on the activation of NF-κB signaling, it may play other critical roles in ATLL pathogenesis. Among the mutations related to TCR/NF-κB signaling, *PLCG1* and *PRKCB* are frequently mutated in ATLL (Table 1). IRF4 plays a major role in NF-κB signaling and genomic alterations, including mutations, have also been identified in ATLL [20]. Several studies have identified the importance of IRF4 in both B-cell lymphomas and PTCL [24,25,26]. Copy number alterations of *CD247* and *DLG1*, both of which are involved in TCR signaling, have also been identified in some ATLL cases.

Although Tax is regarded as a crucial viral product of HTLV-1, silencing of Tax is frequently observed in ATLL [29]. TCR/NF-κB signaling-related genes altered in ATLL are included in the Tax (a HTLV-1 product) interactome, suggesting that mutations can activate TCR instead of Tax for the survival and proliferation of the cells. Alterations of LCK, which is also an important major contributor to TCR signaling in addition to FYN, have not been identified in ATLL. These ATLL cells mainly show CD4 and CD25 expression. LCK assists in the activation of TCR signaling by interacting with the cytoplasmic tail of CD4 or CD8. LCK may play important roles in ATLL pathogenesis as a non-oncogene.

## 4. Immune Escape Mechanisms in ATLL

In addition to several clinical risk factors for ATLL development such as advanced age, the administration of immunosuppressive drugs to HTLV-1 carriers is currently considered a risk factor for early development of ATLL [30,31], suggesting that immune escape mechanisms play important roles in ATLL pathophysiology. Several genomic alterations related to immune escape have been identified in ATLL [19,20,32]. *PD-L1* (*CD274*) alterations are some of the more notable alterations in ATLL [32] and are known to suppress T-cell function via binding to PD-1. Several inhibitors targeting the interaction have shown anti-tumor effects. In addition to the genomic amplification of *PD-L1*, disruption of the three prime untranslated regions (3′-UTR) in *PD-L1* has been identified in 27% of ATLL cases. This disruption stabilizes the mRNA of PD-L1 leading to overexpression, which allows ATLL cells to escape from the host immune response. Furthermore, deletions and mutations of genes related to major histocompatibility complex (MHC) class I, including *HLA-A*,*-B* and *B2M*, have been identified in ATLL cases [20]. CD58 and FAS are involved in the T-cell and NK-cell immune response and both genes are also altered in ATLL [19,20]. In diffuse large B-cell lymphoma (DLBCL), alterations of *B2M* and *CD58* show significant co-occurrence, acting synergistically on the immune escape mechanisms [33]. However, a trend of co-occurrence was not observed in ATLL cases, suggesting the heterogeneous mechanisms related to immune escape in lymphoma. The origin of ATLL is regulatory T-cells (Tregs). ATLL cells suppress surrounding normal T-cell function, similar to Tregs [34]. Therefore, the use of appropriate prophylaxes for infections are highly recommended during treatment of ATLL since patients often develop severe opportunistic infections [35,36,37]. From a genomic standpoint, PD-L1 alterations may also be associated with immunodeficiency in ATLL cases via the suppression of normal T-cells. MYC is involved in the up-regulation of PD-L1 by binding directly to the promoter region of PD-L1 [38]. Since the importance of MYC expression in ATLL cell lines has been confirmed by in vitro analysis [39], the expression of MYC in ATLL may also regulate PD-L1 expression. Although the activation of RAS signaling is also reported to result in the overexpression of PD-L1, the mutation of RAS genes is less common in ATLL than in solid carcinomas [20,22,40]. 

## 5. Heterogeneity of Genomic Alterations in ATLL: Geographic Region and Age at Diagnosis

HTLV-1 is an endemic disease in several countries, including Japan, the Caribbean islands, countries in Middle and South America, Middle East and Australia [41]. While ATLL frequently develops in endemic areas, its incidence has increased in non-endemic areas and decreased in endemic regions [42]. One of the regional variations in ATLL is age at diagnosis. In fact, ATLL cases in South America and the Caribbean islands tend to be younger than ATLL cases in Japan [43,44,45]. Regarding this point, a recent study of clinical characteristics of Caribbean ATLL cases found that median age at presentation was 54 years [44]. In a recent national survey in Japan, the median age was 68 years old [45]. Interestingly, the age at ATLL diagnosis is gradually increasing in Japan, and the same trend may be observed in the Caribbean cases in future. Prognosis of ATLL also differed between Caribbean and Japanese ATLL cases (median survival time; 5.5 months in the Caribbean acute and lymphoma types, 8.3 months in the Japanese acute type, and 10.0 months in the Japanese lymphoma type) [44,45]. This may be related that number of cases with allogeneic SCT were higher in the Japanese cases than in the Caribbean cases. A study analyzed the genomic alterations of ATLL in North America [21] and identified a distinct genomic landscape with frequent mutations related to the epigenome. The *EP300* mutation was found in 20% of North American ATLL cases, compared to 6% of Japanese cases (Table 1). A transcriptional analysis identified distinct transcriptomes between Japanese and North American cases [21], suggesting regional variations in ATLL pathophysiology. Several case reports identified novel mutations of ATLL [46,47]. Since clinical sequencing has been widely adapted, more novel mutations from other endemic areas will be identified through the analysis in future. Genomic alterations of ATLL cases in other regions have not yet been explained but could provide necessary insights into the molecular biology of ATLL. 

In a recent study, we reported on the analysis of genomic alterations in young Japanese ATLL cases [22]. In several types of cancers, young age is associated with unique genomic alterations which result in the development of tumors [40,48,49]. Since fusion proteins induced by structural variations are thought to be involved in carcinogenesis, compared with mutations alone, we screened for genomic fusions and genomic mutations by RNA-sequencing and targeted-sequencing. As a result, both CTLA4–CD28 and ICOS–CD28 fusions were found in 38% of the cases analyzed. Although these fusion genes were previously reported by Kataoka et al. (2015) [20] the frequency of young cases in our study was higher (Figure 1). 

The existence of concurrence in two fusion genes is particularly characteristic of young cases. Using in vitro analysis, we found that both fusion genes transformed normal cells, suggesting that these fusions may be enough to drive ATLL. Our analysis also revealed that the activation of CTLA4-CD28 depended on CTLA4 ligands, which cause CD28 signaling (a key signaling pathway in ATLL). The expression of CD80 and CD86, two CTLA4 ligands, were observed in ATLL cells and tumor-associated stromal cells. Taken together, CTLA4-CD28 acts in both a cell-autonomous and non-cell autonomous manner and is targetable by inhibitors. The potent activity of CTLA4 antibody against CTLA4-CD28 was demonstrated by in vitro assays [22]. The prevalence of these fusions remains unknown, except for Japanese ATLL cases.

## 6. Characteristic Genomic Alterations in ATLL Compared with Other PTCLs

Confirmation of the monoclonal integration of HTLV-1 is needed for ATLL diagnosis. In other words, the morphology of ATLL cells covers all types of PTCLs. Morphological variants of Anaplastic large cell lymphoma-like, Angioimmunoblastic T-cell lymphoma (AITL) and Hodgkin-like lymphoma have been identified in ATLL [50]. Additionally, our group reported that ATLL and a part of PTCL, not otherwise specified (PTCL-NOS), share clinicopathological and genetic characteristics [51,52,53]. The genomic mutations found in ATLL and other PTCLs are summarized in Table 1 [20,27,28]. The *PRKCB* mutation is the second most common ATLL mutation and is relatively specific to ATLL. PRKCB is a protein kinase C (PKC), part of a family of serine/threonine kinases and enzymes that play important roles in several signaling pathways, including TCR and B-cell receptor signaling. Several isotypes of PKC have been identified, with cells predominantly expressing PKC. PKCβ, encoded by *PRKCB*, is mainly expressed on B-cells and mast cells. Therefore, Prkcb-knockout mice have impaired B-cell function after activation by the B-cell receptor [54]. The *PRKCB* mutation is also found in DLBCL, a B-cell lymphoma. However, the frequency is less than 5% among cases [55,56,57]. Therefore, it is interesting that the *PRKCB* mutation is commonly and predominantly observed in ATLL, which is derived from T-cells, after HTLV-1 infection. The *PRKCB* mutation in ATLL has been identified as a gain-of-function mutation and the pathways affected by this mutant play a critical role in the pathophysiology of ATLL [20]. Additionally, mutations of the transcriptional factors GATA3 and IRF4 are predominantly found in ATLL (Table 1). GATA3 is a crucial transcriptional factor for T-cell differentiation and interacts with other transcriptional factors, including ZBTB7B and RUNX1 [58]. Frameshift- and splice-site mutations of GATA3, resulting in intron-retention of GATA3, are frequently observed in ATLL [20]. A recent study on T-cell acute lymphoblastic leukemia (T-ALL) reported that GATA3-driven nucleosome eviction dynamically modulates NOTCH1-MYC enhancer activity and is strictly required for NOTCH1-induced T-ALL initiation and maintenance [59]. Mutant GATA3 in ATLL may also function as an oncogene. However, detailed analyses will be needed in the future. *PLCG1*, *CCR4* and *CARD11* mutations are also found in PTCL-NOS and AITL, as well as in ATLL (Table 1). *STAT3* and *TP53* mutations are common across PTCL subtypes. Although *TET2* and *RHOA* mutations are characteristic to AITL at a high prevalence, these mutants are also observed in some ATLL cases.

## 7. Genomic Alterations in ATLL as Prognostic Markers

Clinical prognostic factors have been identified in ATLL. Classification of the four subtypes in ATLL also depends on clinical findings [6]. Recently, the Japan Clinical Oncology Group’s (JCOG) prognostic index (PI) (JCOG-PI) included Eastern Cooperative Oncology Group performance status and calcium levels. Simplified ATL-PI from age, albumin level, and soluble IL-2 receptor (sIL-2R) level have also been established [60,61]. The level of sIL-2R has been reported as an independent prognostic factor for indolent type [62]. We previously identified copy number alterations (CNAs) of cell cycle-related genes, including *CDKN2A* and *TP53*, and *CD58*, as predictive markers for the transformation of chronic into acute type (Table 2) [19]. Since *CD58* alterations cause immune escape from host immunity, we also analyzed other molecules of immune escape mechanisms and identified the loss of MHC class II expression as an independent poor prognostic factor in the aggressive type [63]. Genomic alterations of *CIITA*, which regulate MHC class II expression, are frequently observed in other types of lymphoma, like Hodgkin lymphoma [64]. Several studies have found that epigenetic mechanisms of the EZH2 mutant suppresses MHC class II expression in acute myeloid leukemia and DLBCL [65,66]. Genomic alterations of MHC class II-related genes and *EZH2* have not been identified in ATLL. However, ATLL cells show high levels of EZH2 expression [67,68]. This EZH2 expression might cause the loss of MHC class II expression, leading to the increased aggressiveness of ATLL. EZH inhibitors targeting EZH2 and EZH1 have proved effective for ATLL [69].

Kataoka and co-workers leveraged their findings of the genomic landscape of ATLL to develop clinicogenetic risk models [70]. In this study, *PRKCB* mutation and *PD-L1* amplification were identified as prognostic factors, in addition to JCOG-PI and age (Table 2), for the aggressive type of ATLL. The mutation of *PRKCB* is characteristic of ATLL, as previously described. Interestingly, the mutation of *PLCG1*, which is involved in the TCR signaling pathway, as well as *PRKCB*, showed no impact on the prognosis, suggesting that PRKCB mutation plays other roles in the cell proliferation of ATLL (Figure 2A). Conversely, *IRF4* mutation, *CDKN2A* deletion and *PD-L1* amplification were significantly associated with poor outcomes among the indolent type. As a result, PD-L1 amplification is regarded as an independent prognostic factor in all ATLL cases. This amplification is associated with escape from the immunosuppression of the surrounding T-cells by binding to PD-1 on the T-cells. As forced expression of PD-L1 on the tumor cells contributed to tumor microenvironment via modulation of angiogenesis [38], such non-cell autonomous interactions may also be involved in the tumor progression of ATLL. As the disruption of 3’UTR of PD-L1 also causes high expression, this alteration may also be associated with the aggressiveness of ATLL. Our immunohistochemical study also revealed that PD-L1 protein expression on ATLL cells was associated with poor prognosis, while the ATLL cases in which tumor-microenvironment cells express PD-L1 showed favorable prognosis in the aggressive type (Figure 2B) [63,71,72]. Further analysis will be required to fully understand the role of PD-L1 expression in the cellular microenvironment and to analyze the clinical significance of PD-L1 in indolent type ATLL at the protein level. One of the important findings of this clinicogenetic model is that the identified genetic model can be used to further classify the chronic type with unfavorable clinical factors [70,73]. Cases with chronic ATLL with unfavorable clinical factors require intensive chemotherapies, including SCT, as in the aggressive type [7]. Observation or the combination of interferon α and zidovudine is recommended for chronic type ATLL cases without any clinical factors. Therefore, intensive chemotherapies may improve the chronic type prognosis with unfavorable factors lacking any genetic risk factors, or cases that do not require intensive chemotherapies. Further analysis will be needed to determine the appropriate therapeutic approaches for chronic type ATLL cases [73].

A recent genomic analysis of HTLV-1 carriers found that those who develop ATLL already carry genomic mutations related to ATLL in their peripheral blood [74]. These alterations may act as predictive markers for ATLL development. Carriers may therefore benefit from early therapeutic treatment.

## 8. Precision Targets from ATLL Genomic Studies

Comprehensive genomic analyses of ATLL have precisely revealed that all identified genomic alterations are subclonal [70,73], indicating that targeted therapies for those genomic alterations may exert partial effects. With this in mind, we will discuss targetable genomic alterations in ATLL.

### 8.1. Mogamulizumab

First, we will discuss the potential use of an anti-CCR4 antibody (mogamulizumab) in ATLL. CCR4 expression is observed in over 90% of ATLL cases and is reported to be associated with a poor prognosis [75]. Approximately 30% of ATLL cases have CCR4 mutations with no differences in terms of prevalence between North American cases analyzed by Nakagawa et al. (2014) and Japanese cases [20,76,77]. The *CCR4* mutation is a gain-of-function mutation that up-regulates its expression. Although we previously identified a *CCR4* frameshift mutation as an independent prognostic factor of ATLL [77], comprehensive subsequent analyses did not find any clinical impact [70]. The activity of mogamulizumab against ATLL cases with *CCR4* mutation was reported by Sakamoto et al. (2018) [78]. In their study, mogamulizumab significantly improved the overall survival of ATLL cases with *CCR4* mutations compared to cases without the mutation. The *CCR4* mutation has previously been regarded as a predictive marker for mogamulizumab response. We found that the level of CCR4 expression was associated with the response to mogamulizumab [79]. Mogamulizumab had no effect on several ATLL cases with the mutant in our cohort, suggesting that further analysis will be required in the future. As neither study evaluated the clonality of the *CCR4* mutation, this may be associated with the response to mogamulizumab. Additionally, our study also found that lymph node lesions had a tendency for resistance to mogamulizumab [79]. This also requires further analysis in the future.

### 8.2. Immune Checkpoint Inhibitors

Overexpression of PD-L1 caused by genomic alterations has been observed in ATLL cases, typically those with poor outcomes. Immune checkpoint inhibitors that target the PD-1/PD-L1 axis may represent an effective treatment for ATLL. However, Ratner et al. (2018 and 2019) reported that ATLL patients showed rapid progression after PD-1 blockade [80,81]. In their analysis, there were no specific genomic alterations related to the progression, suggesting that PD-1 blockade decreased the suppressive roles of the PD-1/PD-L1 axis on ATLL cells. Additionally, T-cell repertoire and HTLV-1 provirus sequencing revealed that HTLV-1 infected immature T-cells in patients progressed after PD-1 blockade. From a transcriptional perspective, ATLL cells that expanded in the peripheral blood after PD-1 blockade were found to express several genes that are characteristic of tumor-associated Tregs. These findings may be associated with the mechanisms of PD-1 blockade resistance in ATLL and further analysis is required to confirm the mechanisms. PD-1 suppresses T-cell function primarily by inactivating CD28 signaling [82]. CD28 alterations, including genomic fusions, are frequently observed in ATLL [20,22]. However, the function of PD-1 in altered CD28 signaling remains unclear but may play an important role in ATLL, as described previously. In addition to PD-L1 expression on the cell surface, exosomal PD-L1 can also directly inhibit T-cell function for several tumors [83]. Exosomal PD-L1 levels distinguished between ATLL cases that responded to PD-1 blockade in metastatic melanoma [84]. Investigating and analyzing these factors may improve our understanding of the mechanisms of PD-1/PD-L1 in ATLL cells. Evaluating the mechanism of the dynamics of immunity is challenging, but ATLL patients will benefit from the corresponding analyses, including single-cell transcriptome analysis, based on clinical trials of immune checkpoint inhibitors.

### 8.3. Lenalidomide

The IRF4/BATF3 transcriptional network also plays a critical role in ATLL pathophysiology [39,85]. As described, genomic alterations of *IRF4* are seen in ATLL, where it acts as a predictive marker for the indolent type. An immunomodulatory agent, lenalidomide, has been reported to target IRF4 in DLBCL and multiple myeloma [25,86]. A phase II clinical trial confirmed the efficiency of lenalidomide in ATLL with an overall response rate of 42% [87]. However, the mechanisms by which lenalidomide suppresses the growth of ATLL cells and the targets of lenalidomide in ATLL cells remain unclear. PTCL cell lines with low IRF4 expression levels were found to be sensitive to lenalidomide, with the levels of IRF4 protein unchanged after lenalidomide administration [26]. Since ATLL cell lines were not included in this study, the efficiency of lenalidomide across ATLL cell lines will need to be evaluated. Since almost all ATLL cells depend on IRF4 for survival [39,85], targeting IRF and including proteolysis-targeting chimeras is a potential therapeutic option for ATLL. The *CSNK1A1* mutation identified in ATLL may be associated with the response to lenalidomide (Table 1). Lenalidomide is highly effective for the treatment of myelodysplastic syndrome (MDS) with deletion of chromosome 5q [88]. Haploinsufficient expression of *CSNK1A1* located on chromosome 5q sensitizes cells to lenalidomide [89]. However, this efficiency would be dependent on the clonality of *CSNK1A1* mutation in each case.

### 8.4. Other Potential Targeted Therapies

Several targeted therapies targeting EZH1/2, Histone deacetylases, DNA methylation, Phosphatidylinositol 3-kinase are currently evaluated in the clinical trials (Table 3). As described above, one of the genomic hallmarks in ATLL is TCR signaling. Among the alterations related to this signaling, PRKCB mutation is the only alteration associated with the prognosis (Figure 2A) A phase I trial using a selective PRKCB inhibitor for chronic lymphocytic leukemia is being conducted (NCT03492125). Although it first needs to check whether the inhibitor can suppress effect of the PRKCB mutant in suitable models, the targeted therapy for PRKCB may be considerable in future. For the mutant proteins, proteolysis targeting chimera (PROTAC) that can induce degradation of targeted proteins may be considerable because several PROTACs were reported to induce both wild type and the mutant proteins [90]. Similar to the PRKCB inhibitor, effect of a MALT1 inhibitor for B-cell lymphoma is also under evaluation (NCT03900598) and it may also be effective for ATLL. Additionally, it has been reported that ALRN-6924, a staple peptide that blocks binding of MDM2 and MDM4 to TP53, was effective for TP53-wild type PTCL cases and acute myeloid leukemia [26,91]. The inhibitor may also have a potent activity to TP53-wild type ATLL cases.

## 9. Targetable Super-Enhancers in ATLL

Super-enhancers, characterized as large clusters of enhancers, are known to play important roles in cancer development, as well as genomic alterations [92,93,94]. These super-enhancers are targetable by several drugs, including CDK7 and bromodomain-and-extra-terminal (BET) inhibitors [95,96,97]. In ATLL, several super-enhancers, including CCR4, TP73, TIAM2, BATF3, MYC and BIRC3, have been identified [39,85,98]. Of these, *CCR4* and *TP73* are also altered at the genomic level, indicating their importance in ATLL. It is of considerable interest to analyze the role of TP73 in ATLL. TP73 has several types of transcriptional variants and its role depends on the type of variant [93]. The *TP73* alteration identified in ATLL may act as an oncogene because it lacks a transactivation domain, which is required for TP73 to act as a tumor suppressor. Further analysis on the association between the transcriptional variants, mutants and super-enhancers of TP73 may tell us more about the pathophysiology of ATLL.

In addition to CCR4, TIAM2 is activated by super-enhancers in ATLL. TIAM2 is a Rac1 selective guanine nucleotide exchange factor that interacts with focal adhesions of cells [94]. Since RHOA, an important gene related to cytokinesis, is also frequently altered in ATLL, this pathway may be crucial in the pathogenesis of ATLL. Interestingly, a small-molecule CDK7 inhibitor called THZ-1 decreased TIAM2 expression in ATLL cell lines, while the knockdown of TIAM2 was found to result in cell death in an in vitro analysis [98]. Although the frequency of ATLL cases with high levels of TIAM2 expression remains unknown, this inhibitor may improve ATLL prognosis. Nakagawa et al. (2018) recently identified a BATF3 super-enhancer in ATLL [39]. It is worth noting that the HTLV-1 virally encoded transcriptional factor HBZ also binds directly to the super-enhancer and regulates the expression of BATF3 and its downstream targets. A BET inhibitor has shown toxicity for ATLL cell lines and patient samples, both in vitro and in vivo. Other super-enhancers, including CD28, FYN and CD2, which are involved in T-cell receptor signaling in ATLL, have been identified with a low frequency [98]. Since the CDK7 and BET inhibitors may also target these super-enhancers, they may be considered as potential therapeutic strategies for use in future clinical trials.

## 10. Conclusions

The comprehensive genomic study of ATLL has allowed precise characterization of its genomic landscape, while functional analyses have led to the discovery of the important roles played by alterations in ATLL. Genomic alterations related to the clinical course of ATLL have also been identified. These analyses have revealed both the inter- and intra-tumoral heterogeneity of ATLL and these findings could be used to improve prognosis. Therefore, clinical trials on the pathological and genomic analysis of this disease can be started. This research should be supplemented and supported by the sharing of findings and issues with ATLL, between clinicians, pathologists and molecular biologists, in the interest of all ATLL patients.

## Figures and Tables

**Figure 1 cancers-13-01801-f001:**
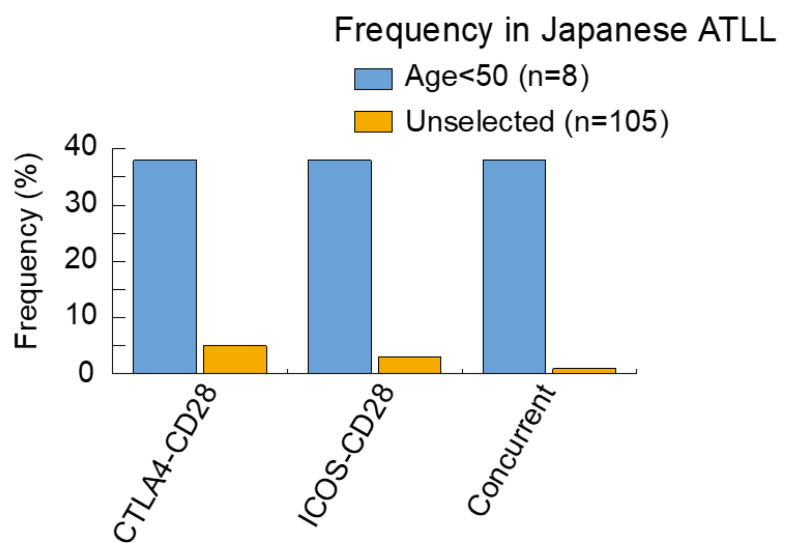
Frequency of CD28 fusions in Japanese ATLL. The frequencies of each alteration are summarized, comparing young ATLL cases < 50 [22] with unselected ATLL cases [20]. Analysis of the unselected ATLL cases may also include young cases.

**Figure 2 cancers-13-01801-f002:**
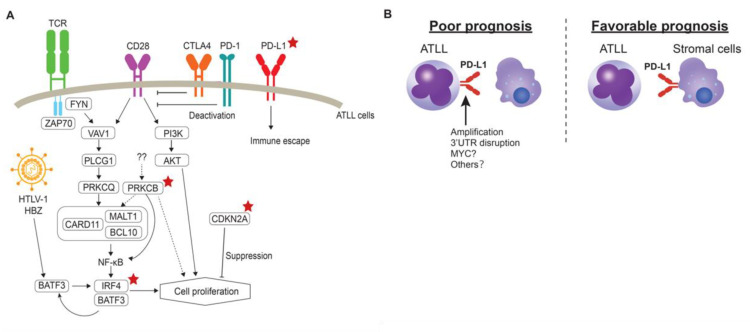
(**A**) Prognostic genomic alterations identified in ATLL. Red stars denote alterations related to prognosis in ATLL. (**B**) Difference of PD-L1 expression patterns on ATLL prognosis. Left: PD-L1 can be expressed through genomic alterations, including the amplification and 3’UTR disruption of *PD-L1* on the surface of ATLL cells. Right: stromal cells in the tumor-microenvironment express PD-L1 and the cases are associated with favorable prognosis.

**Table 1 cancers-13-01801-t001:** Summary of frequencies of genomic mutations commonly found in adult T-cell leukemia/lymphoma (ATLL) and other peripheral T-cell lymphoma (PTCL).

Gene	ATLL Entire Japan [20] (*n* = 370)	ATLL North America [21] (*n* = 30)	ATLL Young Japanese [22] (*n* = 8)	PTCL-NOS [27] (*n* = 133)	AITL [27] (*n* = 26)	ALCL [27] (*n* = 23)	ENKTL [27] (*n* = 25)	HSTL [28] (*n* = 68)
*PLCG1*	36	0	0	8	8	0	0	6
*PRKCB*	33	NE	13	0	0	0	0	NA
*CCR4*	29	NE	38	4	0	0	0	2
*CARD11*	24	7	38	3	0	0	0	2
*STAT3*	21	0	0	5	8	13	12	10
*TP53*	18	23	13	17	0	9	12	10
*VAV1*	18	NE	0	6	8	0	0	NA
*TBL1XR1*	17	13	0	4	0	0	0	NA
*NOTCH1*	15	20	25	5	0	0	0	NA
*GATA3*	15	7	0	0	0	0	0	NA
*IRF4*	14	NE	13	1	0	0	0	16
*FAS*	11	0	0	4	0	4	0	NA
*CCR7*	11	NE	25	2	0	0	0	NA
*POT1*	10	7	0	2	4	4	0	NA
*IRF2BP2*	8	NE	NE	2	0	0	0	NA
*TET2*	8	7	13	43	88	4	0	6
*RHOA*	8	3	13	26	81	0	0	2
*HLA-B*	6	NE	0	5	4	4	0	NA
*HNRNPA2B1*	6	NE	0	0	0	4	0	NA
*EP300*	6	20	0	2	0	0	8	NA
*CD58*	5	NE	0	4	0	9	0	0
*GPR183*	5	NE	NE	1	0	0	0	NA
*CSNK1A1*	5	NE	NE	0	0	0	0	NA
*CSNK2B*	5	NE	NE	1	0	0	0	NA
*CBLB*	4	NE	0	0	0	0	0	NA
*FYN*	4	NE	13	2	0	0	0	3
*B2M*	4	0	0	5	8	0	0	NA
*SETD2*	3	0	0	3	0	0	0	22

ATLL, adult T-cell leukemia/lymphoma; PTCL, peripheral T-cell lymphoma; PTCL-NOS, peripheral T-cell lymphoma, not otherwise specified; AITL, angioimmunoblastic T-cell lymphoma; ALCL, anaplastic large cell lymphoma; ENKTL, extranodal NK/T-cell lymphoma; HSTL, hepatosplenic T-cell lymphoma; NE, not evaluated; NA, not available.

**Table 2 cancers-13-01801-t002:** Genomic alterations related to clinical findings in ATLL.

Alteration	Targets	Clinical Outcome	Reference
CNA of cell cycle-related genes	Chronic type	Early acute transformation	[19]
Del of *CD58*	Chronic type	Early acute transformation	[19]
*PRKCB* mutation	Aggressive type	Poor prognosis	[70]
Amp of *PD-L1*	Aggressive type	Poor prognosis	[70]
*IRF4* mutation	Indolent type	Poor prognosis	[70]
Amp; *PD-L1*	Indolent type	Poor prognosis	[70]
Del; *CDKN2A*	Indolent type	Poor prognosis	[70]

CNA, copy number alteration; Amp, amplification; Del, deletion.

**Table 3 cancers-13-01801-t003:** Potential targeted therapies in ATLL.

Putative Targets	Potential Drugs	Active, or Recruiting Trials Including the Potential Drugs
CCR4	Mogamulizumab	NCT04185220
PD1/PD-L1	Pembrolizumab	
IRF4	Lenalidomide	NCT04301076
	proteolysis targeting chimera	
EZH1/2	Valemetostat Tosylate	NCT04102150
Histone deacetylases	Belinostat	NCT02737046
	Romidepsin	NCT04639843
Phosphatidylinositol 3-kinase	Duvelisib	NCT04639843
CD30	Brentuximab	NCT03113500
	Anti-CD30 CAR-T	NCT04008394
DNA methylation	5-azacitidine	NCT04639843
	OR-2100	
PRKCB	MS-533	
	proteolysis targeting chimera	
MALT1	JNJ-67856633	
MDM2/MDM4	ALRN-6924

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
