# Peer review of "Clinical Applications of Genomic Alterations in ATLL: Predictive Markers and Therapeutic Targets"

_cancers, 2021, doi:10.3390/cancers13081801_

Round 1

Reviewer 1 Report

Overall this is a well-writen and timely article on a devastating disease.

More attention to cases from other parts of the world would be welcome

to improve this review.  For example, a recent north american paper 

reported several novel mutations, which are not referenced in this article

(Current Oncology2021 Feb 6;28(1):818-824. doi: 10.3390/curroncol28010079.).

Reviewer 2 Report

Yoshida, Miyoshi and Ohshima present a thorough but concise review of the genomic aspects of ATLL especially with respect to opportunities for improving diagnosis, prognosis and designing better therapies in these patients.  I thought the text was reasonably well written and informative and this was a good refresher for genomics studies published in the past 5-6 years on ATLL as well as an overview of the disease and role of HTLV-1 and different endemic populations afflicted by this scourge.  It needs only minor revisions and proofreading in order to be published in Cancers

Comments:

  1. From an organizational standpoint I think the manuscript can benefit from having a section regarding an Overview of the genomic landscape of ATLL rather than going right into TCR/NK-kB signaling in section 2 (then more explanation of various features can be explained in other sections). 
  2. There are minor English grammatical errors that may benefit for more editing; the overall reading of the manuscript flows well otherwise.
  3. Table 1 may benefit from a 3d bar graph presentation including potentially more T cell phenotypes (LGL, EATL subtypes, HSTL etc.).
  4. Figure 1 has a comparison from a study of a very small number of patients, if this is to remain then some statistical power/significant description (p value for some frequency estimate for each mutation) should be included and only if significant. Are there other clinical features that can be compared between Japanese and other cases?
  5. Line 56 the percentage of carriers that develop ATLL is mentioned with citations later in the text…I would move that discussion with citation up to this line as I don’t think it is intuitive that the numbers talley in terms of ATLL cases being rare versus ATLL infection is quite common and likely undertested.
  6. Line 200 mentions “Table 2” that does not exist, I think is mistaken for Figure 2. I think a Table with data around prognostic data for genetic alterations would be more useful.  For Figure 2b I don’t quite understand the apparent multiple roles the PD-1/PD-L1 axis is playing a suppressive role…if PD-L1 can be expressed on ATLL cells could it not play a suppressive role too instead of the stromal cells doing that?  I would remove that Figure.
  7. Line 255 “7…targets from molecular biology” is poorly worded as really an target and therapeutic modality is going to be derived from molecular biology. Perhaps the authors intend something like “precision targets from ATLL genomic studies?

Reviewer 3 Report

This manuscript is well documented and organized. However, the potential targeted therapy drugs for ATLL should be discussed and listed in this review.

Round 2

Reviewer 3 Report

The manuscript is well revised according to the suggestions.